# Targeted Environment Design from Offline Data

## Abstract

In reinforcement learning (RL) the use of simulators is ubiquitous, allowing cheaper and safer agent training than training directly in the real target environment. However, this approach relies on the simulator being a sufficiently accurate reflection of the target environment, which is difficult to achieve in practice, resulting in the need to bridge sim2real gap. Accordingly, recent methods have proposed an alternative paradigm, utilizing *offline* datasets from the target environment to train an agent, avoiding online access to either the target or any simulated environment but leading to poor generalization outside the support of the offline data. We propose to combine the two paradigms: offline datasets and synthetic simulators, to reduce the sim2real gap by using limited offline data to train realistic simulators. We formalize our approach as *offline targeted environment design* (OTED), which automatically learns a distribution over simulator parameters to match a provided offline dataset, and then uses the learned simulator to train an RL agent in standard online fashion. We derive an objective for learning the simulator parameters which corresponds to minimizing a divergence between the target offline dataset and the state-action distribution induced by the simulator. We evaluate our method on standard offline RL benchmarks and show that it learns using as few as 5 demonstrations, and yields up to 17 times higher score compared to strong existing offline RL, behavior cloning (BC), and domain randomization baseline, thus successfully leveraging both offline datasets and simulators for better RL.

## 1 Introduction

Recent years have witnessed the proliferation of reinforcement learning (RL) as a way to train agents to solve a variety of tasks, encompassing game playing (Brown & Sandholm, 2019; Silver et al., 2016) and some robotics applications (Akkaya et al., 2019; Schulman et al., 2015; Chiang et al., 2019), among others. These successes often rely on the use simulators, providing several benefits to online RL agent training: large amounts of low cost and more diverse data not readily available in reality (Tolani et al., 2021), and safe training that avoiding physical damage (Krishnan et al., 2021) or human privacy concerns (Tolani et al., 2021). In many instances, for example in game playing, an accurate simulator is easy to devise. In others, rough approximations of the real target environment are sufficient (e.g. Chiang et al. (2019)) to close sim2real gap. However, in other applications such as robot manipulation (Kalashnikov et al., 2018; Todorov et al., 2012), health (Ernst et al., 2006), or web navigation (Gur et al., 2021) designing the simulator to match a desired target environment with sufficient fidelity can be a highly manual process (Nachum et al., 2019a; Krishnan et al., 2021), potentially requiring repeated online access to the target environment (Chebotar et al., 2019; Ramos et al., 2019; Mehta et al., 2020).

Due to the difficulty in matching simulators to target environments and motivated by the successes of supervised learning on large static datasets, a recent line of works Fujimoto et al. (2019); Agarwal et al. (2020); Wu et al. (2019); Levine et al. (2020); Fu et al. (2020); Gulcehre et al. (2020) has proposed to circumvent online training altogether (whether on the simulator or in the target environment) and instead perform RL on an *offline* dataset of logged experience in the target environment, typically collected by an unknown behavior policy. This approach avoids any potential mismatch between training and test environment. However, existing algorithms for offline RL are highly liable to generalization errors, and so these methods often impose a strong regularization on the learned policy to maintain proximity to the offline dataset (Fujimoto et al., 2019; Wu et al., 2019; Kostrikov et al.,

2021; Yu et al., 2020). Thus, such methods can rarely approach the performance of online RL on an accurate simulator.

Here, we propose to leverage both approximate simulators and offline datasets to train agents. Namely, we consider settings in which one has access to an offline dataset collected from a target environment as well an approximate (but nevertheless inaccurate) simulator. How can we leverage both of these to learn a near-optimal policy for the target environment?

To tackle this problem, we propose an approach which, at a high level, is composed of two stages: (1) use the offline dataset to learn a distribution over simulator parameters; (2) train an RL agent in an online fashion on the simulator using any off-the-shelf RL algorithm. The key question is then how to appropriately learn the simulator parameters from an offline dataset, which we term *offline targeted environment design* (OTED). For this problem, we devise an objective quantifying the divergence between the state-action distribution appearing in the offline dataset and the state-action distribution induced by a learned behavior policy acting in the simulator. We show that the simulator parameters and the target behavior policy can be jointly learned such that even a suboptimal behavior policy, that would otherwise perform poorly on the target environment, can help learn simulator parameters very effectively. We further show how the performance of the final returned policy can be improved by employing OTED not only for policy training but also for offline policy *selection* (Yang et al., 2020; Fu et al., 2021). Namely, we show how one may use the OTED-learned simulators to both provide a set of candidate policies as well as evaluate those policies in order to rank and choose the best policy to return, reminiscent of cross-validation techniques popular in supervised learning (Arlot & Celisse, 2010) but so far elusive in RL contexts (Paine et al., 2020).

We apply our method to a variety of domains, including D4RL (Fu et al., 2020) and MiniGrid (Chevalier-Boisvert et al., 2018), showing that our method is able to recover the target environment parameter using as few as 5 demonstrations and learn from high-dimensional image observations. We also show that an online agent trained in designed simulators yields improved performance compared to methods which leverage only simulators or only offline datasets, as well as existing methods for domain transfer (Tobin et al., 2017). On the D4RL (Fu et al., 2020) benchmark, we demonstrate that our method can even outperform an online SAC agent trained on the *ground-truth target* environment. Our method achieves up to 17 times higher score compared to previous state-of-the-art offline RL and behavior cloning (BC) methods as well as domain randomization. On datasets with medium-level behavior that can be collected without domain experts, we show more relative improvements where our model even reaches the performance of the offline RL methods trained on expert-level behaviors.

## 2 RELATED WORK

Our work focuses on learning simulator parameters to match a target environment and is thus related to a large and diverse literature on *domain transfer* in RL. One of the most common and simple approaches to handling unknown simulator parameters is *domain randomization* (Sadeghi & Levine, 2016; Tobin et al., 2017; Matas et al., 2018; Chiang et al., 2019; Tolani et al., 2021; Krishnan et al., 2021), which randomizes the setting of those unknown parameters during training, thus learning an agent which is robust to a wide distribution of possible simulators. While domain randomization has demonstrated successes, the process of choosing appropriate ranges for unknown parameters can be a highly manual process (Andrychowicz et al., 2020), in which one must balance between either too wide range that hampers agent training, and too narrow range that may exclude the target environment parameterization (Nachum et al., 2019a). While some works have proposed heuristic mechanisms to better automate this process (Akkaya et al., 2019; Krishnan et al., 2021), others have explored incorporating online interaction with the environment into this tuning process (Chebotar et al., 2019; Ramos et al., 2019; Mehta et al., 2020). In this work, we avoid any online interactions with the target environment during training, and only assume access to a static offline dataset.

The proposed OTED method, which uses an offline dataset to learn a simulator, is related to recent model-based approaches to offline RL (Matsushima et al., 2020; Yu et al., 2020; Argenson & Dulac-Arnold, 2020). In these works, the offline dataset learns approximate dynamics and reward models of the environment, and the agent optimizes behavior in this approximate model of the environment. Although this makes minimal assumptions on the target environment, using reward and dynamics models as proxies for the environment leads to extrapolation issues which necessitate strong regularizers on the learned policy. Moreover, these existing approaches mostly ignore the

fact that in many scenarios, one already has an approximate simulator, albeit not differentiable; e.g., the physical laws in a continuous control environments are straightforward to implement (Todorov et al., 2012), although parameters more specific to the environment (friction coefficient, actuator gain, gravity) are unknown. Leveraging these simulators and focusing learning on the unknown parameters, as performed by our approach, can thus improve over modelling the full dynamics from scratch.

Our proposed objective is based on a distribution matching loss. Similar losses have appeared in the past, especially in generative models (Goodfellow et al., 2014), off-policy evaluation (Nachum et al., 2019b), off-policy learning (Nachum et al., 2019c), and imitation learning (Ho & Ermon, 2016; Stadie et al., 2017; Kostrikov et al., 2020). These predominantly focus on learning a policy via an adversarial objective and ignore the mismatch between simulators and real environments. Even when the distribution matching objective is optimal, this mismatch might give very suboptimal policies, due to potential poor support of the data distribution. This is a very realistic scenario since tuning simulator parameters accurately is a very tedious process and requires domain-experts.

Finally, we emphasize that our proposed approach is distinct from previous work which suggests to combine offline learning on static datasets with online learning on the target environment (Nair et al., 2020; Ajay et al., 2020; Yang & Nachum, 2021; Xie et al., 2021; Lee et al., 2021). Specifically, we avoid any online interactions with the target environment, and only allow online training on the simulator, which may or may not be a good approximation to the target environment.

## 3 BACKGROUND

We consider an RL setting in which we are given access to both an offline dataset of experience from the target environment and an approximate simulator, in which the agent can gather more experience. We elaborate on the notation and background relevant to this setting.

**Reinforcement Learning** We define an environment as a Markov Decision Process (MDP) given by a tuple $(S, A, \rho_0, T, r, \gamma)$ where $S$ is a state space, $A$ is an action space, $\rho_0$ is an initial state distribution, $T(s'|s, a)$ is a state transition distribution, $r : S \times A \to \mathbb{R}$ is a reward function over state and actions, and $\gamma \in [0, 1)$ is a discount factor. A stationary policy $\pi$ in this environment is a function from states to distributions over actions. In reinforcement learning, we are interested in finding a policy $\pi$ that maximizes the cumulative discounted returns: $\mathbb{E}\left[\sum_{t=0}^{\infty} \gamma^t r(s_t, a_t)|s_0 \sim \rho_0, a_t \sim \pi(s_t), s_{t+1} \sim T(s_t, a_t)\right]$. We will use $d^\pi(s, a)$ to denote the *state-action distribution* of $\pi$: $d^\pi(s, a) := (1 - \gamma) \sum_{t=0}^{\infty} \gamma^t \Pr[s_t = s, a_t = a|s_0 \sim \rho_0, a_t \sim \pi(s_t), s_{t+1} \sim T(s_t, a_t)]$.

**Simulator vs. Target Environment** Typically in online reinforcement learning, a policy is trained based on experience collected by the agent in a *simulator*. While this implicitly assumes that the simulator can perfectly mimic the target environment, in practice this is often not the case. The simulator typically only provides an approximation to the true initial state distribution $\rho_0$ and state transition distribution $T$. For example in robotics applications, while physics simulators can be built, it is infeasible to accurately set all appropriate parameters (e.g., friction coefficients, actuator gains, wind speed). In this work, we assume such a parameterized simulator exists, and it is given by $\mathcal{M}(z) := (S, A, \rho_0(\cdot|z), T(\cdot|\cdot, \cdot, z), r(z), \gamma)$, where $z$ denotes the parameters of the simulated environment. Naive approaches would either set $z$ heuristically or using randomization methods. In this work, we aim to use offline datasets to learn $z$ so that $\mathcal{M}(z)$ approximates the target MDP, which we denote as $\mathcal{M}^*$. We use $d^\pi(s, a|z)$ to denote the state-action distribution of $\pi$ in the parameterized simulator $\mathcal{M}(z)$.

**Offline Reinforcement Learning** Offline RL, as an alternative to online RL on simulators, assumes that an agent cannot collect any online experience and is restricted to a dataset $D$ of $(s, a, s')$ collected by an unknown behavior policy $\mu$; i.e., $(s, a) \sim d^\mu, s' \sim p(s, a)$. We use $d^D(s, a)$ to denote the distribution over state-action pairs appearing in $D$. When trained on a fixed dataset, previous state-of-the-art online reinforcement learning algorithms such as SAC (Haarnoja et al., 2019) underperform due to extrapolation errors (Fujimoto et al., 2019; Kumar et al., 2019), which stem from sparsity of the samples in the dataset. Accordingly, many RL algorithms designed specifically for the offline setting make use of strong regularizations to maintain proximity to the offline dataset (Wu et al., 2019; Kostrikov et al., 2021; Yu et al., 2020).

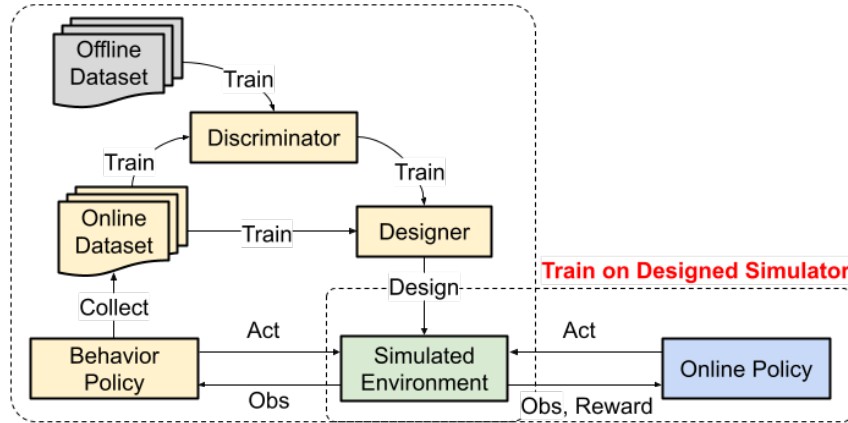

Figure 1: Outline of OTED: The simulator parameters are learned by the *designer*, which uses signals from a learned *discriminator* to match the distribution appearing in the offline dataset to that induced by a learned behavior policy acting in the environment. The learned simulator parameters are then used to train a target policy via standard online RL.

## 4 OTED: OFFLINE TARGETED ENVIRONMENT DESIGN

We now describe our method for leveraging both offline datasets and simulators to learn return-maximizing policies, thus alleviating both of the above issues: (i) data sparsity and extrapolation issues when performing offline learning and (ii) inaccuracy of an approximate simulator for online learning.

At a high-level, our algorithm can be summarized as follows, also illustrated in Figure 1.

1. Learn an approximate behavior policy $\hat{\mu}$ from $D$ using any off-the-shelf imitation learning algorithm; e.g., behavioral cloning (Pomerleau, 1991) or ValueDICE (Kostrikov et al., 2020).

2. Learn a distribution $q(z)$, also called *designer*, over simulator parameters so that $d^{\hat{\mu}}(s,a|z)$ approximates the state-action distribution appearing in $D$.

3. Use any off-the-shelf online RL algorithm – e.g., SAC (Haarnoja et al., 2019) – to learn a return-maximizing policy $\pi$ on the simulator $\mathcal{M}(z)$, where $z$ is sampled anew from the learned distribution $q(z)$ at the beginning of each episode.

Steps (1) and (3) above are straightforward, and so we focus this section on first elaborating on step (2) below, and this is also summarized in Algorithm 1. We then continue to show how we reduce variance inherent in the procedural three-step process above by using replicated experiments, distributions over mixtures of simulator parameters, and model selection without online access.

### 4.1 LEARNING SIMULATOR PARAMETERS VIA DISTRIBUTION MATCHING

Given an approximate behavior policy $\hat{\mu}$, we propose to learn a distribution $q(z)$ over simulator parameters so that $d^{\hat{\mu}}(s,a|z)$ approximates the state-action distribution appearing in $D$. Thus, our objective for $q(z)$ is based on a probability divergence between $d^{\hat{\mu}}(s,a|z)$ and $d^D(s,a)$. To encourage better generalization, we also introduce a prior $p(z)$, and measure a divergence between $d^{\hat{\mu}}(s,a|z)q(z)$ and $d^D(s,a)p(z)$. Our proposed objective is given by,

$$D_{KL}(d^{\hat{\mu}}(s,a|z)q(z)\|d^D(s,a)p(z)) = \mathop{\mathbb{E}}_{\substack{z\sim q(z)\\(s,a)\sim d^{\hat{\mu}}(s,a|z)}}\left[-\log\frac{d^D(s,a)}{d^{\hat{\mu}}(s,a|z)}\right] + D_{KL}(q(z)|p(z)). \quad (1)$$

As desired, it is clear that the first term above encourages the parameter distribution $q(z)$ to match the state-action distribution of the approximate behavior policy $\hat{\mu}$ in the simulator to that of the dataset; meanwhile, the second term regularizes the simulator distribution from collapsing. We can

immediately see that the objective above is an RL (actually, a bandit) objective for $q$, where the actions are $z$, the stochastic reward is given by $\log \frac{d^D(s,a)}{d^{\hat{\mu}}(s,a|z)}$ for $(s,a) \sim d^{\hat{\mu}}(s,a|z)$, and a relative entropy regularizer is applied to $q$. Therefore, one may use any off-the-shelf policy gradient algorithm to learn $q$, as long as one has access to the density ratios $\frac{d^D(s,a)}{d^{\hat{\mu}}(s,a|z)}$.

However, in general one does not have access to the densities $d^{\hat{\mu}}$ or $d^D$ explicitly. Nevertheless, one can easily sample from either density; samples from $d^{\hat{\mu}}$ are given by running the simulator $\mathcal{M}(z)$ and samples from $d^D$ are given by the dataset $D$ itself. A number of algorithms exist for estimating density ratios of two distributions given sampling access. In our implementation, we use the discriminative approach based on (Nguyen et al., 2010). Namely, we train a discriminator $g(s,a)$ to minimize the objective

$$\mathbb{E}_{(s,a)\sim d^{\hat{\mu}}}[-\log(1 - g(s,a))] + \mathbb{E}_{(s,a)\sim d^D}[-\log g(s,a)], \tag{2}$$

similar to the discriminator in GAIL (Ho & Ermon, 2016). When trained to optimality, the discriminator will satisfy the following, $\log \frac{g^*(s,a)}{1-g^*(s,a)} = \log \frac{d^D(s,a)}{d^{\mu}(s,a,z)}$, and this can be used as a reward signal for training $q$.

In practice, to allow for more flexibility in the trade-off between exploitation and regularization, we extend the objective with a tunable parameter $\lambda$:

$$- \mathop{\mathbb{E}}_{\substack{z\sim q(z) \\ (s,a)\sim d^{\hat{\mu}}(s,a|z)}} \left[ \log \frac{d^D(s,a)}{d^{\hat{\mu}}(s,a|z)} \right] + \lambda D_{KL}(q(z)|p(z)). \tag{3}$$

A small $\lambda$ encourages $q$ to exploit the signal from the log-ratios, while a large $\lambda$ encourages $q$ to stay close to the prior $p$. In addition to serving as a trade-off parameter, $\lambda$ can also be interpreted as (i) a reward scaling term in front of log ratios, i.e., $\frac{1}{\lambda} \log \frac{d^D(s,a)}{d^{\mu}(s,a|z)}$ or (ii) a distribution smoothing term applied to both $d^{\mu}$ and $d^D$, i.e., $\log \frac{(d^D(s,a))^{\frac{1}{\lambda}}}{(d^{\mu}(s,a|z))^{\frac{1}{\lambda}}}$.

Similar to (Kostrikov et al., 2019), we use samples stored in a replay buffer rather than online samples to improve sample efficiency. In practice, we adopt the commonly used approximation of log ratios: $-log(1 - g(s,a))$ instead of $-log(1 - g(s,a)) - log(g(s,a))$ in Eq. 3.

---

**Algorithm 1** Training of the simulator parameters and the discriminator

**Input:** Dataset D, initial behavior policy $\hat{\mu}$, initialized discriminator $g$, tunable simulator $S$

1: Gather experience $E$, a sequence of $(s,a)$ pairs, by running the policy $\hat{\mu}$ in the simulator $S$.
2: Estimate discriminator output, $-\log(1 - g(s,a))$ for each sample $(s,a)$ in the experience $E$.
3: Update $q(z)$ using the objective in Eq. 3 with the average $-\log(1 - g(s,a))$ as the reward.
4: Update $g(s,a)$ using the objective in Eq. 2 with mini-batches sampled from $(E, D)$.
5: Repeat above.

**Return:** $(q(z), g(s,a))$

---

**Algorithm 2** Training of a single online agent using off-policy data and multiple simulator models

**Input:** Simulator distributions $q(\cdot|h_i)$ from $n$ different experiments, simulator $S$, empty replay $R$

1: Sample $i \sim \mathcal{U}[1, n]$ and set the simulator $S$ using the mode $z_i$ of the distribution $q(\cdot|h_i)$.
2: Run the online agent $\pi(a|s)$ in the simulator and populate the replay with new experience.
3: Sample a mini-batch from the replay and update the agent parameters.
4: Repeat above.

**Return:** $\pi(a|s)$

---

Our objective, Eq. 1, resembles mutual information when the joint distribution is defined using the proposal distribution and marginal distributions are defined using expert data distribution. The main difference is that we use the behavior policy in the joint distribution and use expert policy in marginals while in mutual information they are the same. Through optimizing the behavior policy to mimic the expert better, the objective will get closer to a mutual information formulation.

## 4.2 REDUCING VARIANCE WITH REPLICATED EXPERIMENTS

Our proposed three step process – approximate $\hat{\mu}$, then learn $q(z)$, and finally training – introduces noise at each step due to randomness in optimizers and learning algorithms. It is important to reduce the variance inherent in this process so that the returned policy is best prepared for evaluation in the target environment. We tackle this problem via two techniques: *simulator selection* and *simulator-based policy selection*.

**Simulator selection** As is typical, we train the objective in Eq. 3 using different hyper parameter settings where each setting will give a different simulator model. Since the target parameter distributions are unknown, this raises the question – how do we choose the ideal set of hyperparameters? We propose to use the first term in Eq. 1, namely expected log ratios, to score and rank candidate simulators trained with different hyperparameters. We detail our procedure to select simulator hyperparameters in below steps:

1. Given a hyperparameter setting $h$, replicate it with $n$ different seeds and train a new OTED for each seed. This will output a design distribution $q(.|h_i)$, a discriminator $g(.|h_i)$, and an approximate behavior policy $\hat{\mu}(.|h_i)$, where we use $h_i$ to denote the use of hyperparameters $h$ with seed $i$.

2. Run approximate behavior policies $\hat{\mu}(.|h_i)$ in their corresponding candidate simulators $\mathcal{M}(z)$ where $z \sim q(.|h_i)$ and collect experience $E$.

3. Estimate discriminator outputs, $c(h_i) = \frac{1}{|E|} \sum_{(s,a) \sim E} -\log(1 - g(s, a|h_i))$ to score $h_i$.

4. Score each hyperparameter $h$ by averaging scores of their corresponding seeds, $c(h) = \frac{1}{n} \sum c(h_i)$.

5. Pick the hyperparameter with the highest score $c(h)$, corresponding to the simulator with the lower approximated KL divergence from the offline data.

**Simulator-based policy selection** Even once a specific simulator is chosen according to the above procedure, training an online RL agent on the simulator can result in drastically different policies, again due to randomness in the learning process. This presents the problem of offine policy selection, which is a well-known challenge for real-world RL (Yang et al., 2020; Paine et al., 2020; Fu et al., 2021). We propose a simple approach to this, described in below steps:

1. Sample design parameters $z_i$ (mode of the $q(.|h_i)$) for each seed from the selected hyperparameter $h$ in the above procedure.

2. Train $m$ seeded on-policy agents $\pi_j$ on the distribution of simulators $\mathcal{M}(z_i)$ (see Algorithm 2).

3. Evaluate the performance of policies $\pi_j$ in each of the simulators $\mathcal{M}(z_i)$.

4. Select the top performing policies according to their achieved the rewards averaged over all simulators.

One can view this offline evaluation as a form of rudimentary cross-validation, similar to existing techniques in offline policy selection which evaluate a learned policy according to $Q$-values learned *separate* from the $Q$-values used to train the policy (Paine et al., 2020). Although the separately-trained $Q$-values (like $\bar{q}(z)$) are trained on the same offline dataset, this simple approach has been demonstrated to achieve strong performance, even compared to more sophisticated methods (Fu et al., 2021).

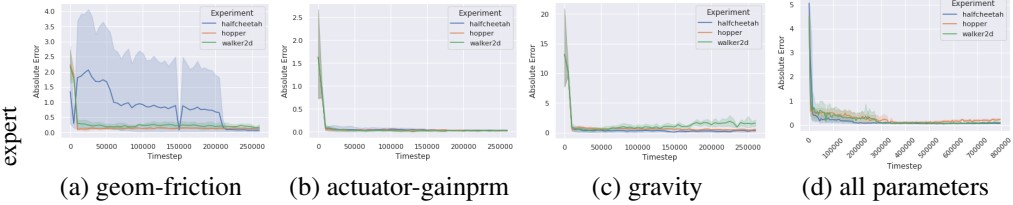

Figure 2: Absolute error of OTED on different simulator parameters using the D4RL expert dataset. We see that OTED is able to accurately learn the true simulator parameters, using only access to a suboptimal offline dataset.

## 5 EXPERIMENTS

### 5.1 EXPERIMENT SETUP

We conduct several key experiments to understand (i) if OTED approximates target design distributions well; if not, what factors contribute to its failure, (ii) if we can select a good simulator model from a large pool of candidates without knowing target design distribution, and (iii) if an online policy trained in a simulator designed by OTED performs well across different environments.

We evaluate OTED, in a variety of settings; different tasks, data sources and simulator parameterizations. We first evaluate our method on 3 **continuous control** tasks from the Mujoco (Todorov et al., 2012) simulator, *HalfCheetah, Hopper,* and *Walker2d*, against state-of-the-art offline RL methods on the previous D4RL benchmark (Fu et al., 2020). D4RL datasets are divided into 3 levels, *medium*, *medium-expert*, and *expert*, reflecting the competency of the behavior policy used to collect the data. We modify the Mujoco simulator so that three 1-dimensional physics parameters – *geom-friction*, *actuator-gainprm*, and *gravity* – that cause different effects on the underlying simulator physics are unknown (See Appendix B for their ground truth values). We learn them both independently and jointly. Next, we experiment with a **discrete** MiniGrid (Chevalier-Boisvert et al., 2018) environment to study if OTED can also learn from high-dimensional image observations. We parameterize a MiniGrid environment by the *density of obstacles* in a 10-by-10 grid.

We use SAC as the online agent specifically as it is not designed to work in the offline RL setting and call an online SAC agent trained in designed environments, OTED-SAC. We compare its performance to (i) previous state-of-the-art offline RL methods, behavioral cloning (BC), F-BRC (Kostrikov et al., 2021), MBOP (Argenson & Dulac-Arnold, 2020), (ii) online SAC agents trained with domain randomization (DR) (Tobin et al., 2017) and a baseline called interval search (IS) that is inspired by ADR ? where parameters are sampled from a search interval that is gradually expanded over time based on the performance of an agent in the simulator, and (iii) a hybrid BC algorithm trained with both provided offline data and simulated data where the latter is collected using DR (BC+DR). We report *(mean) absolute error* to measure the error of final design predictions and ground truth designs. We also report *online reward* in ground truth target environments for OTED-SAC and above baselines.

We compare our model selection method to 2 baseline scoring metrics: (i) *design deviation*, standard deviation of final design predictions across different seeds and (ii) *learning progress*, average log ratios collected during training. We pick simulator models with the lowest score for the former and highest for the latter. We report two metrics to evaluate model selection: (i) *design error*, difference between a design prediction and the ground truth design, and (ii) *design regret*, difference between the design errors of a design prediction and the best prediction possible from the pool of available candidates.

We give more details on our model implementation in the Appendix A and will open source our code to public upon publication. We also present experiments on alternative datasets taken from Kostrikov et al. (2020) in the Appendix C.

## 5.2 RESULTS ON D4RL BENCHMARK

We present our results in three stages, first focusing on the ability of our distribution matching objective to recover the true simulator parameters (Section 5.2.1), then presenting the results of the final learned policy according to OTED-SAC (Section 5.2.2), and finally performing an ablation highlighting the importance of our model selection procedure (Section 5.2.3).

### 5.2.1 ACCURACY OF SIMULATOR PARAMETER PREDICTIONS

In Figure 2, we evaluate OTED on D4RL expert dataset using absolute error metric. We observe that OTED exhibits low error in learning every parameter both independently or jointly. This shows that OTED is capable of learning when parameters can interact in unknown and complicated ways. As an interesting use case, HalfCheetah emits a very high variance on the *geom-friction* parameter early on but converges to a smaller error. This is due to one experiment giving a much higher error and taking much longer to find the correct parameter rather than sensitivity of the HalfCheetah environment for this parameter. Upon inspection all parameter learning experiments, we observe that *geom-friction* is consistently the hardest parameter to learn and is the most prone to trapping a simulator model in a local optima. One particular reason is that, usually, the other two parameters are learned very quickly, allowing the discriminator to return high log ratios even before *geom-friction* is learned. In some experiments, this leads to a simulator model following the inaccurate discriminator and converging into sub-optimal solutions. But, thanks to our simulator selection, we mitigate this issue and train online agents with accurate simulators.

We present more results with all expert levels in Appendix D.

### 5.2.2 PERFORMANCE OF THE ONLINE AGENT WITH DESIGNED SIMULATORS

As OTED exhibits low absolute error on average, the natural question to ask is – does this correlate well with the performance of an online agent trained on designed simulators? In Table 1, we compare OTED-SAC to baseline RL algorithms using different expert levels. 100% performance in D4RL benchmark corresponds to a SAC agent trained in the target environment. For medium-level expert datasets, OTED-SAC clearly outperforms previous RL algorithms. This is particularly important as collecting medium-level demonstrations can be done effortlessly without an optimal policy. While previous algorithms fail when the offline dataset is collected via a sub-optimal policy, OTED is still able to recover the unknown design distribution. Note that we also use BC to induce a behavior policy which is very sub-optimal across all medium-level datasets. Even though Walker2D environments exhibit higher variance on *geom-friction* parameter for medium and medium-expert level experts, OTED-SAC outperforms previous models and also performs better compared to other parameterizations. In majority of the cases, OTED-SAC achieves more than 100% performance, outperforming a baseline SAC agent trained on the ground-truth target environment. Using offline and simulated data in BC algorithm (BC+DR), we get better performance than BC in medium and medium-expert datasets, particularly for Walker2D environments. Our method still outperforms this strong baseline, showing improvements beyond combining offline and simulated datasets.

### 5.2.3 SIMULATOR SELECTION

As presented above, success of an online agent is closely related to a simulator being an accurate reflection of the target environment. As a result, it is very critical to pick a simulator model from a large pool of candidates. In Table 2, we compare our simulator selection method to other baseline methods. Our proposed method outperforms baselines and picks the best model available in almost of cases (zero regret). Interestingly, design deviation is the second best metric and usually has low regret. We hypothesize the success of this metric to its ability to distinguish models with a number of different sub-optimal predictions; hence high deviation from the average prediction.

### 5.3 MINIGRID ENVIRONMENT RESULTS

We now move on MiniGrid, which presents us with image-based tasks. We focus our evaluation on the simulator learned by OTED, showing that OTED is able to recover the true simulator parameters, even in image-based environments.

| | BC | F-BRC | MBOP | DR | IS | BC+DR | OTED-SAC geom friction | actuator gainprm | gravity | all params |
|---|---|---|---|---|---|---|---|---|---|---|
| halfcheetah-medium | 36.1 | 41.3 | 44.6 | 6.7 | 12.2 | 36.6 (0.1) | 97.0 (1.6) | 106.7 (0.9) | **118.8 (2.2)** | 116.6 (2.4) |
| halfcheetah-medium-expert | 35.8 | 93.3 | 105.9 | 6.7 | 12.2 | 50.6 (4.6) | 80.0 (5.6) | 109.3 (2.3) | 113.6 (1.4) | **114.2 (3.3)** |
| halfcheetah-expert | 107.0 | 108.4 | - | 6.7 | 12.2 | 105.7 (1.0) | **119.0 (1.5)** | 109.6 (3.8) | 110.5 (4.6) | 116.3 (1.4) |
| hopper-medium | 29.0 | 99.4 | 48.8 | 73.9 | 20.9 | 30.0 (0.1) | 107.3 (5.8) | 94.2 (8.6) | 80.4 (9.9) | **109.8 (2.6)** |
| hopper-medium-expert | 111.9 | 112.4 | 55.1 | 73.9 | 20.9 | 53.0 (13.3) | **114.0 (0.6)** | 98.5 (4.2) | 109.6 (1.0) | 81.9 (10.5) |
| hopper-expert | 109.0 | 112.3 | - | 73.9 | 20.9 | 111.4 (0.4) | **113.8 (1.1)** | 100.4 (5.0) | 72.3 (18.8) | 45.2 (4.9) |
| walker2d-medium | 6.6 | 78.8 | 41.0 | 46.2 | 26.9 | 16.0 (1.0) | 113.4 (4.1) | 103.3 (12.9) | 115.8 (3.7) | **126.3 (3.6)** |
| walker2d-medium-expert | 11.3 | 105.2 | 70.2 | 46.2 | 26.9 | 13.0 (3.2) | 110.4 (1.9) | 60.1 (18.6) | 128.8 (2.9) | **121.3 (2.8)** |
| walker2d-expert | **125.7** | 103.0 | - | 46.2 | 26.9 | 45.6 (6.9) | 119.7 (2.7) | 102.8 (3.7) | 110.0 (1.8) | 102.6 (2.5) |

Table 1: Performance comparison of our work (OTED-SAC) to existing RL algorithms on D4RL benchmark. We report offline RL algorithm results from (Kostrikov et al., 2021). OTED-SAC is able to recover a near-expert policy across almost all settings. For *geom-friction*, we achieve on-par or better than previous models. *actuator-gainprm* and *gravity* gives slightly lower performance than learning *geom-friction*. We see almost no performance drop when all parameters are trained jointly in medium-level data.

| | Design Deviation | Learning Progress | Log Ratios (OTED) |
|---|---|---|---|
| halfcheetah-expert | **0.064 (0.003)** | **0.064 (0.003)** | **0.064 (0.003)** |
| halfcheetah-medium-expert | **0.061 (0.016)** | **0.061 (0.016)** | **0.061 (0.016)** |
| halfcheetah-medium | 0.073 (0.005) | **0.071 (0.007)** | **0.071 (0.007)** |
| hopper-expert | 0.419 (0.024) | **0.196 (0.03)** | **0.196 (0.03)** |
| hopper-medium-expert | **0.158 (0.044)** | 0.621 (0.455) | 0.621 (0.455) |
| hopper-medium | **0.203 (0.025)** | 0.235 (0.034) | **0.203 (0.025)** |
| walker2d-expert | 0.127 (0.021) | 2.607 (0.856) | **0.101 (0.02)** |
| walker2d-medium-expert | **0.144 (0.03)** | 0.421 (0.141) | **0.144 (0.03)** |
| walker2d-medium | 0.37 (0.037) | 1.416 (0.667) | **0.287 (0.038)** |

Table 2: We ablate over different mechanisms for hyperparameter selection for the learned simulator. We present the design errors. We find that OTED's use of expected log ratios for hyperparameter selection performs best.

We show our MiniGrid results in Figure 3 where we present absolute error between learned parameter and ground truth parameter. After being trained for only several hundred steps, the discriminator learns to distinguish between generated and expert data, emitting a very strong signal for the designer to learn the simulator parameter. Note that $0.26$ is not in the support set of the parameter distribution as we quantize by increments of $0.05$ but OTED is still able to find the optimal prediction. This shows (i) the capability of OTED to generalize to observations that are not seen during training and (ii) the robustness to quantization errors. This also explains why it takes slightly longer for the discriminator to learn.

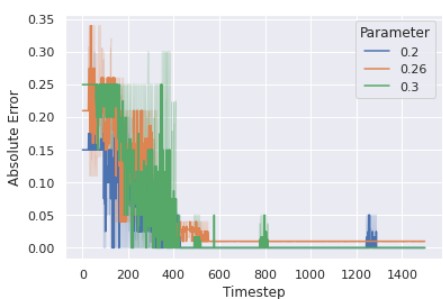

Figure 3: Absolute design error of OTED using different target design parameters in MiniGrid.

## 6 CONCLUSION

We introduced a principled method, OTED, to learn simulator parameters of a target environment from offline datasets, combining two different paradigms, namely offline and online RL. We proposed a KL-divergence regularized distribution matching objective where a distribution over simulator parameters is learned to minimize the discrepancy between the offline dataset and state-action distribution of an online behavior policy in the induced simulator. We also introduced unsupervised simulator and

policy selection methods. By jointly learning the behavior policy and design distribution, we showed that OTED is able to recover the unknown parameter with high accuracy. When evaluated on previous benchmarks, including D4RL, OTED exhibited low absolute error on average. The proposed model selection method consistently outperforms baselines and leads to successful online agent training. An online SAC agent (OTED-SAC) trained with designed simulators performed on par or better than previous state-of-the-art offline and online RL methods. While it is still an open question what is an ideal simulator interface, we believe this depends on the problem domain. For example, friction is an important parameter in robotics which depends on the type of environment such as water or air. We also would like to emphasize that any available parameter can be made explicit and learned using offline data. We showed that using some domain knowledge to set some of the parameters and reduce parameter space is plausible and can give competitive results while reducing complexity. As future directions, we are looking forward to testing OTED in more complex environments such as UI navigation with real human traces.

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
