# OpenReview forum: "Targeted Environment Design from Offline Data"
_ICLR.cc/2022/Conference — ICLR 2022 Submitted_

### Official Review · Reviewer_huR6 · 2021-10-25

**Correctness:** 3
**Technical Novelty And Significance:** 3
**Empirical Novelty And Significance:** 2
**Recommendation:** 6
**Confidence:** 4

**Details Of Ethics Concerns:**

No concerns.

**Main Review:**

# Strengths

1) Novel approach
	- Offline RL methods have generally focused exclusively on the policy learning process. Shifting focus to the environment parameterization is a valuable change in approach. Previous efforts in this vein have been limited to efforts like domain randomization that do not attempt to learn a good simulator for the target task.
2) Promising simulator fitting results
	- Results are promising for the degree of accuracy in the simulator fits. There is clearly room to improve when allowing fits of many parameters, but the initial results are encouraging.


# Weaknesses

1) Modest policy performance
	- Given the amount of tuning of the simulator and policy selection processes, the results in Table 1 show very modest results. Behavior cloning seems to generally be a viable alternative to the simulator in most cases and is substantially easier to implement and train.
		- Action: Explain why BC would offer such strong performance.
		- Action: Perhaps add an explanation of the additional benefit to having a simulator vs only the final policy.
	- Table 1 does not seem to fully support the claim "We see almost no performance drop when all parameters are trained jointly": both hopper-medium-expert (~20 points lower) and hopper-expert (30 points lower than the next worst) values seem much lower than the single parameter variants.
		- Action: Could this be explained in terms of the environment dynamics? (Or connected to the results in Appendix D with difficulty of learning the hopper environment?)
2) Lack of error bars / statistical tests for policy training results
	- The core results on policy training (Table 1) have no measure of variance or a statistical test of the differences among methods.
	- I recognize the method is computationally intensive to train, but these results are weakened by the lack of any statistical validation of the outcomes. Repeated trials are necessary to establish confidence in the outcomes.
	- I highlight this outcome as it is the core end goal of using simulator tuning: obtaining a better policy for the real task.
	- Action: Add some form of repeated trials and statistical differences test to the policy training results.
3) Lack of statistical tests of differences for the simulator ablations
	- Action: Add repeated trials and statistical tests to show differences (or lack thereof) among the ablations of OTED.

Note: The abstract mentions strong performance with as few as 5 demonstrations. I could not find where these results were reported.

Minor note: I would suggest choosing a different name for the parameter $\epsilon$ as this is typically the symbol used to indicate a noise parameter for sampling distributions (or a very small scalar quantity). For example, $\lambda$ is a widely used name for a trade-off weighting parameter.


**Summary Of The Paper:**

The paper introduces an approach to tune a parameterized simulator to produce data samples that mimic a given offline dataset of trajectories from real data. The core technical contributions involve defining a KL divergence to quantify the difference of the simulator data to a given offline dataset, a training algorithm that interleaves simulator tuning and agent training on simulator data, and selection algorithms to choose a hyperparameterization among simulators and a policy among a set trained on a simulator. Evaluations assess how well the training procedure can recover known simulator parameters and the performance of policies trained on a learned simulator to policies trained with offline RL algorithms on the offline data directly. Results show the simulator can accurately recover parameters and is able to obtain similar performance to offline RL algorithms when tuning certain parameters.

**Summary Of The Review:**

The paper presents an important approach to address offline RL training by tuning a parameterized simulator to produce training data for a policy. This has great potential for further investigation and comparison/combination with other offline RL approaches. The empirical results are weak in the current form, leaving me less certain the work is mature enough for publication. I incline toward seeing this as sufficient contribution to merit acceptance.

My score would improve if the authors can provide clear statistical evidence of the superiority of the new method.

---

> ### Author Response · Authors · 2021-11-20
> **Response to reviewer huR6**
>
> Thank you for your insightful feedback. We will address each of your comments below.
>
> **Why does BC perform well?**
>
> When there is sufficiently large and clean data, BC learns good policies. This is closer to the supervised learning setting where using the traditional cross-entropy loss leads to consistently good results. However, when the data is poor, such as medium-level data in D4RL benchmark, its performance drops significantly and it can’t generalize beyond the data as also shown by other prior work (Kostrikov et al. 2019).
>
> **Simulator vs final policy.**
>
> We think an accurate simulator has benefits beyond a final policy. An accurate simulator can be useful to solve other tasks without collecting new data for each task separately. For example, by learning the simulator parameters using a single offline data, we can assign different goals to an agent in the simulator and learn broader navigation behaviors that could otherwise be infeasible to learn from only a single limited data. We hope that this would also help sample efficient policy training and show benefits beyond offline RL methods such as BC.
>
> **All params performance in Table-1.**
>
> Thank you for pointing this out. We meant to compare the performance on medium-level data. We clarified our paper to reflect this.
>
> **Error bars and repeated experimentation.**
>
> As we explained in the general response, we used 5 seeds to train OTED and 15 seeds to train online policies. We added missing error bars in our figures and tables including our new experiments.
>
> **Number of demonstrations.**
>
> We used as low as 5 demonstrations in ValueDice experiments that we report in Appendix C. For D4RL, we use all available demonstrations that the benchmark provides.
>
> **Notations.**
>
> We updated \epsilon to \lambda in our paper.

---

### Official Review · Reviewer_2itJ · 2021-10-30

**Correctness:** 4
**Technical Novelty And Significance:** 2
**Empirical Novelty And Significance:** 2
**Recommendation:** 3
**Confidence:** 5

**Main Review:**

1.	Strengths and Weaknesses

\+ The idea of using an offline dataset to calibrate the simulator is intuitive and interesting. This is an alternative way of using the offline datasets to help the training of decision-making agents.

\- The proposed method is relatively heuristic and may have limitations on the novelty. As the authors mentioned, using the policy gradient method to optimize simulator parameters could have a large variance. The proposed solution (picking the best hyperparameter after multiple fittings) in this paper may not be applicable in high-dimensional cases due to the high computational cost.

\- If I understand correctly, one assumption of the proposed method is that the model of the simulator is correct and only the simulator parameters are inaccurate. If so, this would limit the application of this method in more complex tasks, where we are unable to access or build a correct model. I suspect that the Offline RL method may outperform the proposed method.

\- There are lots of components in multiple stages in the proposed method. However, the analyses of these components are not enough. For example, the authors use a behavior cloning (BC) method as a surrogate model to collect data from the simulator. Why do they select BC methods and why is this a good choice? What if the BC model is not well-trained and makes the learning of the simulator fail?

2.	Questions

(1)	What if we don’t have an accurate model of the simulator? For example, there could be some friction parameters that are not considered in simple simulators. In this case, the performance of the proposed method will always have a gap. In some worst cases, the learned simulator is too simple to fit the offline data and the online agents overfit the simulator. Could the author provide more experiments on an inaccurate simulator model with fewer parameters than the testing environment?

(2)	The joint training of the discriminator and parameter generator could be unstable. For example, a strong discriminator may make the generator fail. How do the authors deal with this problem?

(3)	In equation (3), a hyperparameter $\epsilon$ is introduced and the authors say that it can balance exploitation and safety. What does safety mean here? I assume this paper is not related to safety since this term is only mentioned here.

(4)	What’s the influence of the seed $i$ in Section 4.2? How large is $i$ enough to reduce the high variance? Are there any theoretical or empirical analyses about it?


**Summary Of The Paper:**

Motivated by the gap between training environments and testing environments, this paper proposes to use offline data to calibrate the simulator then use online RL methods to train generalizable agents. To minimize the divergence between offline data and simulator parameters, the authors train a discriminator. They also propose two cross-validation methods to select the best simulator parameters to increase the performance.



**Summary Of The Review:**

In general, this paper proposes a very interesting direction, but important analyses and experiments are missing in the current version. I tend to reject this paper before the discussion based on my concerns in the main review. I may raise the score if the authors can provide reasonable answers to my questions.

---

> ### Author Response · Authors · 2021-11-20
> **Response to reviewer 2itJ**
>
> Thank you for your detailed feedback. We will address your concerns below including extensive new experiments that we also added to our paper.
>
> **Novelty and model details.**
>
> We would like to highlight the contributions of our work: (i) We introduced a principled objective for inducing simulator parameters based on a KL-regularized distribution matching, (ii) We showed that it is capable of learning accurate simulator parameters in continuous and discrete control tasks, and (iii) Through extensive experimentation, we examined various use cases and illustrated limitations of our models. We believe that our work would foster further research in this area.
>
> Instead of using policy gradient, we practiced an alternative formulation of our objective similar to the soft-actor critic (see Appendix A for more details) to reduce the variance.
>
> One of our goals in discussing hyperparameter selection was the lack of a standardized way in the RL community. Previous works typically report magic numbers without explaining how to choose hyperparameters or they use the performance of an agent in the test environment to choose a hyperparameter. Whereas, in supervised learning, hyperparameters are chosen using a development dataset separate from the test data. We proposed a simple but effective solution for choosing hyperparameters for simulator tuning without evaluating our agents in the ground truth environment. We report the performance of the chosen hyperparameter setting using 5 seeds.
>
> **Misspecified simulators.**
>
> We ran experiments [Exp-2] to understand what happens if one of the simulator parameters (gravity) is fixed to an incorrect value and OTED is trained to learn the remaining two parameters. We observe that OTED’s performance drops as the error in the fixed parameter increases. But, OTED can be robust up to a certain error level. For example, in HalfCheetah, we see that 50\% error in the fixed parameters leads to almost no drop in performance. For other environments, we see a similar trend for medium-level data.
>
> **Choice of BC for learning the behavior policy.**
>
> We explain the reason why we chose BC for the D4RL dataset in the general response (we use ValueDice for additional experiments in Appendix C).
>
> **Inaccurate simulator.**
>
> We argue that even though there can be more realistic simulators, there will always be gaps when compared to reality. And in the simulator design, the main question is how to create the cheapest simulator that trains the agent sufficiently well to minimize the sim2real gap. We believe our model is a way to make it easier to create better and more realistic simulators by automating parameter tuning with any offline data.
> In addition to our findings above (“Misspecified simulators”), we also ran additional experiments [Exp-1 and Exp-3] to understand the effect of the gap between a simulator and real environments. We used two different settings: noisy offline data and unknown (but correctly set) simulator parameters. We observe that OTED’s performance degrades with increasing noise levels in the offline data. But, it can be robust up to certain noise levels. For example, in HalfChetah and Hopper, up to 0.5 standard deviation leads to almost no drop in performance. When only 2 of the parameters are learned and the remaining parameter is set to the ground truth value, we get similar performance as learning all parameters.
>
> **Instability.**
>
> Diversity of data in the initial replay and \epsilon (updated to \lambda in the paper) in the KL-divergence term were important for the model to learn well (see Appendix A for model details). We observed that if diversity of data was poor in the initial replay, the discriminator could overfit and the generator would converge to local optima. We solve this problem by truncating initial episodes earlier (100 steps as opposed to 1000 steps) and keeping the number of initial replay the same. Since initial episodes are sampled from a simulator with random parameters, this gives much more diverse data and leads to better discriminator training. For the \epsilon term, we observe that as the first term in Eq. 3 gets smaller, KL-term becomes more dominant and the designer is forced to match the prior more closely; which can lead to instability. We solved this problem by decaying the epsilon term over time so that as designer policy is more accurate, the model is regularized less.
>
> **Safety.**
>
> Thanks for pointing this out. Our work is not related to safety and we used safety as an additional intuition to using a KL regularizer where a higher \epsilon (updated to \lambda in the paper) can encourage the designer to stay closer to the prior. We replaced safety with regularization in our paper.

---

> > ### Author Response · Authors · 2021-11-20
> > **Response to reviewer 2itJ**
> >
> > **Influence of seeds.**
> >
> > We ran additional experiments [Exp-4] where we range the number of seeds from 5 to 25 and report online policy results. We observe that using more than 5 seeds gives better results but no visible difference between 10 or more seeds. We also observe that our model selection approach leads to consistently better results compared to just aggregating all available seeds. To clarify, take an experiment with 10 seeds. After training 10 online policies in parallel, we use our approach in Section 4.2 to select top-5 seeds and report aggregated results of these five. Alternative to this is just aggregating results of all 10 seeds which we show to underperform.

---

> > > ### Comment · Reviewer_2itJ · 2021-11-21
> > > **Response to authors**
> > >
> > > Thank the authors for addressing my concerns and providing so many new experimental results. After carefully reading the response and opinion from other reviewers, I still feel this paper have two main problems.
> > >
> > > **1. The contribution**
> > >
> > > I totally agree with the authors and other reviewers that the targeted problem in this paper is very interesting. However, the proposed solution in this paper is basically a black-box optimization method (because the simulator is non-differentiable) consisting of several existing components that are widely used in GAN and RL. As a black-box optimization method, the biggest problem is the learning efficiency since we can only get information from samples. I cannot see intuitive modifications in this method that are novel to increase efficiency. This is the main reason I feel the novelty of this paper is not enough: it is more like a combination of existing methods to solve a new and interesting problem. I think being a tier 1 ML conference, ICLR should require more novelty than this. (This is just the bar in my mind, the final criterion is decided by AC).
> > >
> > > As for the replicated experiments, in the response, the authors explain that some RL algorithms miss this part. I cannot agree with this point since it is a fair game for all methods to select their own best hyperparameters. Hyperparameter searching is a standard way in the machine learning area thus it is definitely not a contribution (unless anyone proposes a more efficient way). Also, I don’t think this is a novel way to reduce variance since you spend much more time just by getting more samples with Monto Carlo. However, I am not saying the author should not do this, but I don’t think it is necessary to use a whole page to discuss this. I think it is indeed an interesting direction to think about how to reduce variance in this simulator modeling setting, but it should at least have some sort of mathematical analysis.
> > >
> > > **2. The misspecified simulator**
> > >
> > > I am not convinced by the response from the author about this problem. The authors provide some new experimental results and say:
> > >
> > > > But, OTED can be robust up to a certain error level. For example, in HalfCheetah, OTED still gives good performance even when there is a 50% error in the parameter (using -4.0 vs the ground truth -9.81)
> > >
> > > This empirical robustness varies from case to case and I don’t think this error level bound is convincing. But at least we have a sense of how large the influence of a wrong simulator model is. The authors only test one inaccurate parameter, but I guess the influence is more serious when there are more inaccurate parameters and even missing parameters. This is definitely the biggest limitation of the proposed method. Especially, the initial motivation of this paper is to use offline data that is usually collected from real-world or complex systems. I agree that this is a really hard problem and I don’t have specific answers. Maybe more empirical experiment results could be a way to make this method convincing when we cannot reach any theoretical guarantees. Again, I totally agree this is an interesting but hard problem. Any attempts to solve this misspecified simulator problem are exciting. However, I don’t think the current version of this paper works in this direction enough. Whether this paper is accepted or not, I encourage the authors to explore more in this direction.
> > >
> > > Based on the above two points, I tend to keep my score and reject the current version of this paper.

---

> > > > ### Author Response · Authors · 2021-11-22
> > > > **Response to reviewer 2itJ**
> > > >
> > > > Thank you for your feedback. We will address your comments below.
> > > >
> > > > **Contribution (1st paragraph)**
> > > >
> > > > We agree with the reviewer that our model is “a combination of existing methods to solve a new and interesting problem.” We show that this simple approach can get surprisingly good results, although at the same time it has certain limitations. Our extensive experimentation highlights a variety of future research directions to circumvent these limitations.
> > > >
> > > >
> > > > **Contribution (2nd paragraph)**
> > > >
> > > > While hyperparameter searching is standard and necessary in ML, the key issue we highlight is the reliance of previous offline RL methods on the use of the target environment itself for hyperparameter tuning, which contradicts the initial motivations for offline RL to avoid online access to the target environment [1]. Our paper’s contribution is significant for illuminating how learning an approximate simulator can avoid this issue.
> > > >
> > > > **Mis-specified simulator**
> > > >
> > > > Any sim2real method will inevitably degrade in performance as the gap between simulation and target environment increases. Our extensive empirical results show that OTED is no exception to this. As the reviewer mentions, our results show exactly the point at which OTED is no longer robust. We believe any good paper should show such limitations clearly and accurately.
> > > >
> > > > **Sim2Real**
> > > >
> > > > We are also working on improving OTED’s performance with real offline data. We hope to provide additional insights into our problem and bridge the sim2real gap in a more realistic setting. We will share our findings.
> > > >
> > > > [1] https://arxiv.org/abs/2007.09055

---

### Official Review · Reviewer_mcpN · 2021-11-02

**Correctness:** 3
**Technical Novelty And Significance:** 3
**Empirical Novelty And Significance:** 3
**Recommendation:** 5
**Confidence:** 4

**Main Review:**

### Strengths
- The proposed method for learning simulator parameters seems well motivated and supported by compelling empirical evaluations.
- Performance comparisons seem thorough and the method seems especially helpful with lower quality demonstrations, since it only uses the suboptimal demonstrations for simulator parametrization and not policy learning.  Overall, the idea of using an offline dataset of suboptimal demonstrations to learn simulator parameters for online learning is interesting.

### Weaknesses
- The abstract mentions using this method to close the sim2real gap, however it is hard to say how well this will extrapolate to sim2real applications, as this relies on the offline dataset and online learning domains to be identically parametrized. i.e. even if we have an offline dataset of real world demonstrations, if our simulator can’t isolate and identify the correct parameters to adapt (which is likely much harder with the sim2real domain gap compared to the simulated gap here), would the method still work?
- The design deviation method also seems comparable to OTED with respect to design errors (in Table 2) — are there evaluations on task performance with this scheme?
- Missing standard deviations to results presented in Tables 1, 2.

### Questions
- How sensitive is the parameter learning to the approximate behavior policy learned (i.e. is a reasonably good policy necessary or will a random policy be sufficient for learning simulator parameters)?

### Additional Feedback
- Would be interesting to include an ‘oracle’ method — online SAC with ground truth parameters to see the gap with OTED-SAC, if any.

**Summary Of The Paper:**

The authors propose a method, Offline Targeted Environment Design, OTED, for using offline datasets to train ‘realistic’ simulators by minimizing the divergence between the state-action distributions of the offline dataset and in the learned simulator. The authors also introduce methods for selecting learned simulators and policies across simulators, and compare their method against a range of offline RL methods and online RL methods with variations of domain randomization.

**Summary Of The Review:**

The method is intuitive and it seems like it is capable of learning the chosen simulator parameters well, potentially leading to novel methods in combining system identification and offline RL. However, it is unclear how necessary the more complex sampling method is (compared to design deviation), and applicability to real world system identification problems seems limited. The paper would be strengthened by examining how well the method works in adapting 'realistic' simulators from real world offline data, where there is more likely to be interactions between multiple simulator parameters and the domain gap is larger.

---

> ### Author Response · Authors · 2021-11-20
> **Response to reviewer mcpN**
>
> Thank you for your feedback and insights into the paper. We will address your concerns below and updated our paper.
>
> **Sim2Real and simulator specification.**
>
> We focus on the incorrect transition dynamics aspect of the sim2real gap as we explained in the general response. But we also ran additional experiments [Exp-1 and Exp-2] to understand other aspects of sim2real as well.
> We studied a simulated sim2real gap by adding zero mean gaussian noise with varying standard deviations to the offline observations [Exp-1]. We observe that OTED’s performance drops with increasing noise. But, OTED can also be robust up to certain error levels. For example, in HalfCheetah and Hopper, we see no visible drop in performance even using 0.5 standard deviation.
> We also ran more experiments [Exp-2] where the simulator is misspecified and one of the parameters (gravity) is fixed to an incorrect value. We make the same observation. OTED’s performance drops as the error in the fixed parameter increases. But OTED can still be robust up to a certain error. For example, in HalfCheetah, even just using half of the ground truth gravity (-9.81), OTED still gives accurate simulator models.
> The main reason for drop in performance in both cases can be explained by discriminator overfitting more when offline data is far from online data.
> We believe research in designing better simulator interfaces and learning simulator parameters are interesting and complementary research directions.
>
> **Experiments with design deviation.**
>
> We explained our findings in the general response [Exp-6].
>
> **Accuracy w.r.t behavior policy.**
>
> Using a good behavior policy can give better results but it is not necessary. In fact, we use the same behavioral cloning (BC) method that is used to get baseline results in D4RL benchmark (Table 1). BC performs poorly when using suboptimal data (all medium-level data in Table-1). We observe that OTED is still able to predict simulator parameters with low error (Table 2) using the same poor BC model.
>
> **Oracle model.**
>
> 100% in D4RL benchmark corresponds to a SAC agent trained in the ground truth target environment. We clarified this in Section 5.2.2.

---

> > ### Comment · Reviewer_mcpN · 2021-11-22
> > **Response to Authors**
> >
> > First, I would like to thank the authors for their thorough response and additional experiments, and I agree with the other reviewers that the problem setting is very interesting and important.
> >
> > My main concern is still regarding using this method for reducing the sim2real gap, where oftentimes the simulator is unlikely to be perfectly specified. While the additional experiments in Appendix E and F are helpful, I am not entirely convinced that OTED is robust to misspecified simulator models -- e.g. the error is only low for HalfCheetah in Figure 10. This can be concerning as the investigations from Figure 11 show that increased error in the simulator parameters can cause large amounts of policy performance degradation. Overall, the empirical results seem to indicate the method is limited to the case when the offline data comes from a very similar model to the simulator, with only very few known parameters to learn. As such, I am inclined to keep my current score.

---

> > > ### Author Response · Authors · 2021-11-22
> > > **Response to reviewer mcpN**
> > >
> > > Thank you for your feedback.
> > >
> > > Any sim-to-real method will inevitably degrade in performance as the gap between simulation and target environment increases. Our extensive empirical results show that OTED is no exception to this. As the reviewer mentions, our results show exactly the point at which OTED is no longer robust. We believe any good paper should show such limitations clearly and accurately.
> > >
> > > We are also working on improving OTED’s performance with real offline data. We hope to provide additional insights into our problem and bridge the sim2real gap in a more realistic setting. We will share our findings.

---

### Official Review · Reviewer_rXE7 · 2021-11-02

**Correctness:** 3
**Technical Novelty And Significance:** 3
**Empirical Novelty And Significance:** 3
**Recommendation:** 8
**Confidence:** 4

**Main Review:**

### Strengths

Overall, I think this paper could be a valuable contribution at ICLR 2022 since it is one of the first to make a reasonably convincing case for the effectiveness of combining offline and online RL. Moreover, it shows how one can leverage (a limited amount of) prior knowledge about the environment to create an accurate simulator, while inferring from data the parts which are unknown / uncertain. I believe is a suitable setting for many problems of interest.

The paper is clearly written, technically correct, the proposed approach is novel (as far as I know) and evaluated quite extensively on a variety of tasks, and compared with strong baselines and ablations.

### Weaknesses

I think the paper could use some improvements and there are some missing experiments / analysis that could make it stronger, as detailed below:

1. It looks like you are missing the results on MiniGrid. Can you please include and describe them in the rebuttal? It would be useful to know how well OTED performs when learning from images in discrete domains, in order to fully evaluate the contribution and the generality of the method across different benchmarks.

2. Is there a reason for which you are not comparing with adaptive domain randomization (ADR; OpenAI et al. 2019) or your DR baseline essentially ADR? Can you please include more details about the DR baseline and explain how it differs from ADR or add the comparisons with ADR?

3. How does performance change with the accuracy of the learned simulator parameters? This could shed light into the robustness of the method with respect to errors in the learned simulators. This could also help better understand the trade-offs between the compute needed to learn the simulator and the performance gains.

4. I think the paper should lay out more transparently the kinds of assumptions made regarding the simulator. It should also include a discussion regarding deviations from these assumptions such as large number of parameters that need to be inferred or the possibility of having a wrong model of the simulator. How could one discover the parameters that need to be inferred or detect model misspecifications? I think more context could help readers to understand when this method is suitable and when it is not and make it clear that it does use some prior knowledge about MDP.

5. The paper is also missing a discussion of the limitations of the proposed approach and future directions stemming from these ideas. These are important to place the work in the larger research context and inspire follow-up contributions.

6. Another concern I have is regarding how the performance (and simulator accuracy) scale with the number (or range) of the parameters you need to learn. For example, if you have less prior knowledge about the simulator, you may need to infer multiple parameters. I think an experiment showing how this scales would provide additional insight.

7. The related work section seems to be missing some relevant work such as "AWAC: Accelerating Online Reinforcement
Learning with Offline Datasets" by Nair et al. 2021. This paper also uses a combination of online and offline RL (although it assumes limited access to the true simulator), so a discussion of the two is needed.

### Minor

1. Do you have an intuition regarding why there is divergence in Figure 2 for gravity in walker and for all params in hopper?
2. Can you bold the model with the best performance in the tables for readability?
3. Please proof read. There are a few typos on pages 1, 6, 8, and likely more.




**Summary Of The Paper:**

This paper proposes a new approach, which combines offline reinforcement learning with learning in simulation, without the need of training on the target environment. More specifically, they propose to use offline data to learn a distribution over the simulator's parameters and then use the inferred simulator to train an RL policy online. The experiments show that the proposed approach, OTED, performs significantly better than other offline RL, behavior cloning, and domain randomization methods, particularly when learning from non-expert demonstrations as well as in the low-data regime.

**Summary Of The Review:**

Given all the above, I am currently leaning towards recommending acceptance. If the authors address the above concerns (adding MiniGrid experiments, ADR comparisons, analysis of how the performance scales with the simulator's accuracy, and a discussion of the method's limitations), I plan to recommend acceptance (unless the other reviewers change my mind or realize I have missed some crucial details).

---

> ### Author Response · Authors · 2021-11-20
> **Response to reviewer rXE7**
>
> Thank you for your comments; we appreciate the detailed feedback and updated our paper to reflect your suggestions and new experiments.
>
> **Additional results for minigrid.**
>
> In addition to the design error of OTED in Figure-3, we also added a plot for KL-divergence between behavior policy and ground truth target policy in Appendix K. We observe that OTED is able to find the optimal simulator parameter (Figure-3) and the behavior policy closely approximates the ground truth policy (Appendix K).
>
>
> **ADR baseline.**
>
> We have a baseline named interval search (IS) that is essentially the same as ADR. The main difference is that instead of tuning the entropy of the distribution, we tune the interval of a uniform distribution. We start with a very narrow interval around a random center and gradually increase the width of the interval as long as the behavior policy is improving. We observe that this leads to poor performance. We clarified these in our paper.
>
> **Online policy performance w.r.t simulator errors.**
>
> To measure the effect of the accuracy of simulator parameters, we ran additional experiments [Exp-5]. We observed that even small errors in simulator parameters can lead to drops in performance. This is more exacerbated if errors are relatively larger.
>
> **Simulator misspecification and interface.**
>
> We ran additional experiments to understand how misspecifications in the simulator can affect OTED’s performance [Exp-2]. We observe that if there are misspecified and hidden simulator parameters, OTED’s performance can drop. But we also observe that OTED can be robust up to a certain error in hidden parameters. For example, in HalfCheetah, we don’t see a visible drop in performance even with 50\% error in the parameter. While it is still an open question what is an ideal simulator interface, we believe this depends on the problem domain. For example, friction is an important parameter in robotics which depends on the type of environment such as water or air. We also would like to emphasize that any available parameter can be made explicit and learned using offline data. We showed that using some domain knowledge to set some of the parameters and reduce parameter space is plausible and can give competitive results while reducing complexity. We also added these discussions to the conclusion.
>
> **Limitations of OTED.**
>
> OTED has several limitations: (i) A simulator needs to have an interface with a sufficient coverage over parameters, (ii) online data in the simulator should be diverse enough for the discriminator to train well, and (iii) complexity of the parameter space can adversely affect OTED’s performance. In our experiments [Exp-2], we showed that setting hidden parameters incorrectly can degrade OTED’s performance. Our initial empirical examinations also showed that if online data is not diverse enough, the discriminator overfits which gives poor log ratio estimations. We increase diversity of online episodes by truncating them early while keeping the size of the replay the same (see Appendix A for more details). As future directions, we are looking forward to testing OTED in more complex environments such as UI navigation with real human traces. We added these discussions to Appendix L and conclusion.
>
> **Performance w.r.t number of parameters to learn.**
>
> In addition to the experiments with a misspecified simulator [Exp-2], we also experimented [Exp-3] with learning only 2 of the parameters while keeping the other parameter fixed. We observe similar results as learning all of the parameters. One interesting observation is that OTED exhibits high error when the gravity parameter is learned. The reason is that while the relative error of OTED is still low w.r.t the gravity parameter, its absolute error is high. This is because the gravity has a relatively high scale (-9,81) where even a relatively small error can give high absolute error.
>
> **Related Work.**
>
> Thanks for pointing out this work. We added a comparison in the related work section.
>
> **Divergence of parameters.**
>
> One particular reason is that as the designer policy improves, the first term in Eq. 3 becomes relatively small and KL-term dominates. This forces the designer to match the prior more closely which leads to some divergence. We also explain how we alleviate this problem in Appendix A.

---

> > ### Comment · Reviewer_rXE7 · 2021-11-21
> > **Thank you**
> >
> > Thank you for the comprehensive response.
> >
> > I think the empirical evaluation has been significantly improved in light of the reviewers' feedback. I appreciate the authors' efforts in running additional experiments, particularly the ones studying the effects of a misspecified simulator, the errors in the learned parameters, and the number of parameters that need to be inferred. I believe these provide more insight into the strengths and weaknesses of this approach.
> >
> > The answer on ADR wasn't completely convincing to me. While I understand that your IS baseline is similar to ADR and I think that is a good comparison to include in the paper, I still think it would be valuable to compare your approach directly to ADR, as that instantiation has been shown to be very effective in these types of settings (i.e. the adaptation of the parameters' distribution and not merely their range). I encourage the authors to include comparisons with ADR in the next draft.
> >
> > In conclusion, I recommend acceptance as the paper proposes a new approach for an important problem, and the analysis of this method is quite thorough.

---

> > > ### Author Response · Authors · 2021-11-22
> > > **Response to reviewer rXE7**
> > >
> > > Thank you for your feedback and increasing your score. We really appreciate it. We will compare our method with ADR and add related results to the final version of our paper.

---

### Author Response · Authors · 2021-11-20
**General Response.**

We thank all the reviewers for their thorough feedback. We addressed some of the common comments here as well as in the individual comments.  We believe our findings will foster further research on training better simulator models from offline data. We list a summary of changes that we made to address the comments.

**Why did we choose BC to learn the behavior policy?**

We chose BC as it is simple to implement and its poor performance for some datasets (especially medium-level data in D4RL) allowed us to analyze OTED’s performance with a suboptimal behavior policy. Our results in Table-1 and Table-2 show that BC can indeed be poor but OTED is still able to learn an accurate model of the simulator parameters. We also point out that we have additional results using another dataset in the Appendix C where we used a state-of-the-art offline RL method, ValueDice (Kostrikov et al., 2019), to learn the behavior policy. OTED was able to learn an accurate simulator in this case as well.

**Repeated experimentation and error bars.**

In our paper, we used 5 seeds to train OTED and 15 seeds to train online policies. We added standard error to every table in the paper and made sure every figure also includes error bars. The same is true for our new experiments as well.

**Clarification on sim2real.**

We would like to clarify that our focus in sim2real gap is in making transition dynamics as close to the target environment as possible. Even when the simulated observations are as realistic as possible, incorrect transition dynamics can lead to poor results. Offline RL benchmarks and mujoco control suite allowed us to examine this gap in isolation and understand how an accurate simulator can help alleviate this gap. To understand other aspects of the sim2real gap, we also performed additional experiments with noisy offline observations and misspecified simulators as we detail below.


**New experiments that we performed to understand various different aspects of our problem.**

* *[Exp-1] How would OTED perform in a setting where offline observations are far from simulated observations?*
We conducted controlled experiments to understand how the performance of OTED changes w.r.t an increasing gap between offline and online data. We used additive gaussian noise with varying standard deviation to modify the offline data and retrained OTED. As expected, OTED’s performance drops with higher noise but OTED can still be robust up to a certain noise, especially for HalfCheetah and Hopper environments. The reason for drop in performance is that as offline data becomes less similar to online data, the discriminator overfits which leads to poor log ratio estimations. See Appendix E for more details.

* *[Exp-2] How would OTED perform in a misspecified simulator?*
To understand OTED’s performance in this setting, we fixed the gravity parameter to an incorrect value and let OTED learn the other two. We used different values for gravity to measure the performance w.r.t different error levels. As expected, the performance of OTED drops as the gravity parameter becomes more inaccurate. The reason is similar to [Exp-1] where the discriminator overfits. But, OTED can be robust up to a certain error level. For example, in HalfCheetah, OTED still gives good performance even when there is a 50\% error in the parameter (using -4.0 vs the ground truth -9.81). We think that these experiments also shed light on how OTED would perform in more realistic settings. See Appendix F for more details.

* *[Exp-3] What is OTED’s performance for learning 2 parameters? More specifically, using geom_friction & actuator_prm, geom_friction & gravity, and actuator_prm & gravity.*
We ran additional experiments where only 2 of 3 parameters are learned while the remaining is assumed to be correct. We observe a similar trend where OTED performs similarly to learning all three parameters. We observe higher errors with gravity involved. The main reason is that gravity has relatively high ground truth value where a relatively small error can still give high absolute error and bias the results. See Appendix G for more details.

* *[Exp-4] What is the effect of using a larger number of seeds when doing model selection in online policy training?*
We experimented with using [5, 10, 15, 20, 25] seeds to train an online agent and compared results using both with and without model selection. We observe that using more than 5 seeds can be beneficial but we don’t see a considerable difference when using more than 10 seeds. We also observe that our model selection method outperforms just aggregating all available seeds. See Appendix H for more details.

---

> ### Author Response · Authors · 2021-11-20
> **General Response.**
>
> * *[Exp-5] How sensitive is the online policy w.r.t errors in simulator parameters?*
> For this experiment, we sampled simulator parameters from a manually curated grid and trained online policy for every parameter combination. We show that even small inaccuracies in the simulator can give a considerable drop in online policy performance. This motivates the need for tuning simulator parameters for better policy performance. See Appendix I for more details.
>
> * *[Exp-6] What is the performance of using design deviation instead of log ratios for selecting OTED?*
> We would like to first clarify that all model selection criteria in Table-2 are introduced by our work and they are not alternatives to OTED but merely different ways of choosing a designer policy. All these criteria require a well trained OTED and they can be collected with no additional cost.
> We ran experiments where we chose designers using design deviation criteria instead of log ratios. We observe similar performance between the two. But, we want to point out a significant drawback of the design deviation; it is agnostic to any shifts in design distribution. We give more details related to this in the Appendix J.

---

### Decision · Program_Chairs · 2022-01-20

**Decision:**

Reject

**Comment:**

This paper proposes a new approach, which combines offline reinforcement learning with learning in simulation. There were different views on the paper among the reviewers and we had quite a lot of discussions. As a consequence, there were still serious concerns remaining, e.g., whether the results are significant enough, whether there are clear advantages of the proposed mehtod over directly using offline RL methods. It is not justified whether the proposed framework can use offline data more efficiently or better reduce the gap between mismatched simulators and offline data. The reviewer who gave the highest score decided not to champion the paper. Considering all the discussions, we believe the paper is not ready for publication at ICIR yet.